# Sublethal Concentrations of 2C-I and 25I-NBOMe Designer Drugs Impact *Caenorhabditis elegans* Development and Reproductive Behavior

**DOI:** 10.3390/ijms26073039

**Published:** 2025-03-26

**Authors:** Eva Gil-Martins, Daniel José Barbosa, Fernando Cagide, Fernando Remião, Fernanda Borges, Renata Silva

**Affiliations:** 1Associate Laboratory i4HB-Institute for Health and Bioeconomy, Faculty of Pharmacy, University of Porto, 4050-313 Porto, Portugal; evagilmartins18@gmail.com (E.G.-M.); remiao@ff.up.pt (F.R.); 2UCIBIO-Applied Molecular Biosciences Unit, Laboratory of Toxicology, Department of Biological Sciences, Faculty of Pharmacy, University of Porto, 4050-313 Porto, Portugal; 3CIQUP-IMS/Department of Chemistry and Biochemistry, Faculty of Sciences, University of Porto, 4169-007 Porto, Portugal; fernando.fagin@fc.up.pt; 4Associate Laboratory i4HB-Institute for Health and Bioeconomy, University Institute of Health Sciences-CESPU, 4585-116 Gandra, Portugal; daniel.barbosa@iucs.cespu.pt; 5UCIBIO-Applied Molecular Biosciences Unit, Translational Toxicology Research Laboratory, University Institute of Health Sciences (1H-TOXRUN, IUCS-CESPU), 4585-116 Gandra, Portugal

**Keywords:** designer drugs, 2C-I, 25I-NBOMe, *Caenorhabditis elegans*, toxicity

## Abstract

Designer drugs like 2C-I and 25I-NBOMe have emerged as potent psychoactive substances, with several reports linking their consumption to severe poisoning and fatalities. However, there is limited information on their toxicity, particularly in in vivo models. In this manuscript, we evaluate the survival, developmental, and reproductive impact of these designer drugs on the model organism *Caenorhabditis elegans* (*C. elegans*). For this purpose, adult worms synchronized at the L1 stage were exposed to growing concentrations of 2C-I and 25I-NBOMe. The animal survival rate and the putative effects of the drugs on *C. elegans* development and reproductive behavior were assessed after 24 h of exposure. A concentration-dependent decrease in animal survival was observed. 25I-NBOMe was approximately six times more toxic than 2C-I (LC_50_ values—1.368 mM for 2C-I and 0.236 mM for 25I-NBOMe). Furthermore, sublethal concentrations of both drugs delayed animal development and reduced the total progeny but not its survival. Overall, these findings underscore the developmental and reproductive risks associated with exposure to 2C-I and 25I-NBOMe, even at sublethal concentrations.

## 1. Introduction

The consumption of designer drugs, also known as new psychoactive substances (NPSs), has raised significant public health concerns due to their unpredictable effects and potential toxicity. Designed to mimic the effects of traditional illicit substances, their widespread availability, ease of access, and low cost have driven their growing popularity [1,2]. Among the many NPSs currently available on the drug market, phenethylamine derivatives, including 2C-I and 25I-NBOMe (Figure 1), have garnered considerable attention.

2C-I, or 4-iodo-2,5-dimethoxyphenethylamine (Figure 1, left), belongs to the 2C family of phenethylamines, a class of compounds first synthesized by Alexander Shulgin in the late 20th century [3]. On the other hand, 25I-NBOMe, or 2-(4-iodo-2,5-dimethoxyphenyl)-*N*-(2-methoxybenzyl)ethanamine (Figure 1, right), is a derivative of 2C-I and part of the NBOMe series, a class of compounds known for their exceptional potency as serotonin receptor agonists, particularly at the 5-HT_2A_ receptor [4].

Interestingly, both 2C-I and 25I-NBOMe induce 5-HT_2A_-dependent behaviors in rodents, such as wet dog shakes (WDSs), back muscle contractions (BMCs), and head twitch responses (HTRs) [5,6,7,8]. Moreover, administration of 25I-NBOMe (0.3 mg/kg, IP) has been shown to induce conditioned place preference (CPP) in C57BL/6J mice, suggesting a rewarding effect [9]. Although 25I-NBOMe does not interfere with self-administration parameters (number of infusions and active/inactive lever presses), it significantly increases dopamine levels in the striatum and cortex of rodents [7,9].

Currently, 2C-I and 25I-NBOMe are used recreationally for their psychedelic effects. Despite being advertised as safer alternatives to classical psychedelics, these substances have been linked to several acute adverse effects and fatalities [10,11,12,13,14,15,16,17]. Evidence from case reports and in vitro experiments indicates that NBOMe drugs have a higher binding affinity to serotonin receptors and exhibit increased toxicity compared to their 2C counterparts [18,19,20]. Recently, we explored the cytotoxic effects of 2C-I and 25I-NBOMe using differentiated SH-SY5Y cells and primary rat cortical cultures. Our results revealed that 25I-NBOMe is significantly more cytotoxic than 2C-I. For both drugs, we observed clear signs of mitochondrial dysfunction, including increased intracellular Ca^2^⁺ levels (only 25I-NBOMe), decreased intracellular ATP levels, and mitochondrial membrane depolarization, though there were no significant changes in intracellular ROS levels. Additionally, both substances induced apoptosis, but interestingly, only 2C-I was found to induce autophagy and strongly trigger caspase-3 activation [21]. However, to date, there are no in vivo studies assessing the impact of 2C-I and 25I-NBOMe on development and reproductive behavior, which is vital for understanding their broader biological effects and potential risks. Building on these findings, our current study transitioned to the *Caenorhabditis elegans* (*C. elegans*) in vivo model to further investigate the toxic effects of these drugs, including on animal development and reproductive behavior. Its small size (about 1 mm for adults) and rapid life cycle (approximately 3 days) make it an ideal model for studies of this nature. Moreover, *C. elegans* shares many genes and signaling pathways with mammals, and its transparent body and measurable behaviors allow real-time observation of drug effects, providing valuable insights into human–drug action and supporting the extrapolation of results to the human context [22,23].

## 2. Results

### 2.1. Impact of 2C-I and 25I-NBOMe on C. elegans Survival

2C-I and 25I-NBOMe significantly affect, in a concentration-dependent way, the *C. elegans* survival rate, with 25I-NBOMe being significantly more toxic than 2C-I (Figure 2 and Appendix A). Notably, 25I-NBOMe was approximately six times more toxic than 2C-I, as indicated by their LC_50_ values of 0.236 mM and 1.368 mM, respectively. The higher toxicity of 25I-NBOMe aligns with its previously predicted higher lipophilicity, which is thought to increase intracellular permeation (Figure 2), as well as with the in vitro experiments in cultured neuronal cells that compared 25I-NBOMe to its 2C-I counterpart [21].

### 2.2. Impact of 2C-I and 25I-NBOMe on C. elegans Development over Time

To evaluate the potential interference of 2C-I and 25I-NBOMe on animal development, we measured the animal length, as it correlates well with the developmental stage [24]. The animal body length was significantly affected by exposure to the drugs 2C-I and 25I-NBOMe, with a concentration-dependent reduction in worm length observed at both 24 and 48 h after the 24-h exposure to the selected drug concentrations (Figure 3b).

For 2C-I, after 24 h, worm length significantly decreased by approximately 15% at the 0.5 mM concentration and by approximately 25% at the 1.0 mM concentration. This trend persisted after 48 h, where similar results were observed (Figure 3b). For 25I-NBOMe, the effects were milder but still significant. At 24 h, the worm length significantly decreased by approximately 10% at 0.05 mM and by approximately 15% at 0.10 mM. After 48 h, similar results were observed, although this effect was not significant at the lowest concentration tested (Figure 3b). These results indicate that both substances induce significant and visible developmental changes in *C. elegans* even at sublethal concentrations (Figure 3c,d, Table 1).

### 2.3. Impact of 2C-I and 25I-NBOMe on C. elegans Reproductive Behavior

In addition to animal development, we also explored the putative effects of 2C-I and 25I-NBOMe drugs on animal reproductive behavior. Exposure to 2C-I and 25I-NBOMe for 24 h significantly reduced the animals’ brood size (F1 generation) in a concentration-dependent manner (Figure 4b). For 2C-I, the tested concentrations of 0.5 and 1 mM significantly reduced the animals’ brood size by approximately 15 and 25%, respectively. For 25I-NBOMe, only the higher, sublethal concentration of 0.1 mM resulted in a significant reduction in the animal’s brood size, by approximately 20% (Figure 4b). However, the reduction in the total progeny (Figure 4b) was not accompanied by significant alterations in the percentage of hatched embryos (Figure 4c), suggesting that these drugs impair animal fertility, but not the viability of their progeny. Moreover, a morphological examination of the animals’ vulval region was conducted to identify potential abnormalities that could be connected to reproductive health and behavior. However, as observed in Figure 4d, drug-treated worms show no visible vulva abnormalities.

## 3. Discussion

2C-I and 25I-NBOMe are synthetic phenethylamines that have gained widespread attention as recreational drugs due to their potent psychedelic properties. However, their unregulated status and potential to cause severe adverse effects raise serious concerns for public health. Despite their popularity, much remains unknown about their toxicological profiles, including their mechanisms of action and the full range of risks associated with their use. Investigating these toxic effects is critical for understanding the risks to users and for effective regulation. In vivo models, such as *C. elegans*, offer a promising and cost-effective alternative for assessing the toxicity of drugs like 2C-I and 25I-NBOMe. With a short life cycle, well-defined biology, and genetic tractability, *C. elegans* provides practical advantages over traditional rodent models and serves as a valuable tool for toxicity assessments relevant to human biology. As a whole-organism model, *C. elegans* allows the observation of systemic effects and interactions within a living system, bridging the gap between in vitro and mammalian models [22,25,26,27,28].

To the best of our knowledge, this study is the first to use *C. elegans* to assess the toxicity of 2C-I and 25I-NBOMe, including their effects on development and reproductive behavior. Our results demonstrated that both drugs exhibit concentration-dependent toxicity, with 25I-NBOMe being significantly more toxic than 2C-I. This aligns with several studies reporting that the presence of the *N*-2-methoxybenzyl group (in NBOMe drugs) significantly increases the cytotoxicity of 2C drugs, which is consistent with the higher lipophilicity of NBOMe drugs compared to 2C drugs [18,19,21]. In agreement with these findings, studies using other animal models, such as zebrafish, support the high toxicity of NBOMe drugs [29,30,31].

Recently, our group reported that 25I-NBOMe was approximately three times more cytotoxic than 2C-I towards neuronal cells. Both compounds showed concentration-dependent cytotoxicity in differentiated SH-SY5Y cells and primary rat cortical cultures, with EC_50_ values of around 35 μM for 25I-NBOMe and 120 μM for 2C-I [21]. This indicates that cell culture models are, as expected, more sensitive than the *C. elegans* animal model. The lower sensitivity observed in *C. elegans* may be attributed to its complex biological systems, which influence drug absorption, distribution, metabolism, and excretion, resulting in a more moderated cytotoxic response. This lower sensitivity mimics protective mechanisms found in higher organisms, allowing *C. elegans* to reflect more accurately the complex interplay of biological processes that influence drug toxicity in humans, including real-world exposure scenarios. In contrast, in vitro cell models provide direct drug exposure to cells without these mitigating effects, resulting in increased sensitivity to cytotoxic effects [22,32]. Thus, while both models are valuable for toxicological research, the *C. elegans* model offers distinctive advantages for predicting and understanding drug effects in complex biological systems.

Understanding the effects of drugs on developing organisms and reproductive health is essential, particularly considering the prevalence of substance use among young adults. Young adults, particularly those under 35, are the most common users of NPSs, with those aged 18 to 25 representing about 4 in 10 users [33]. According to the 2019 European School Survey Project on Alcohol and Other Drugs (ESPAD), an average of 3.4% of students had tried a NPS during their lifetime, and 2.5% had used them in the past 12 months, reflecting higher usage rates than for amphetamine, MDMA, cocaine, or LSD individually [34]. Furthermore, research shows that the brain continues to develop into young adulthood [35], making this period particularly vulnerable to drug exposure, which may result in development impairments, cognitive deficits, increased risk of psychopathology, or potential substance use disorders later in life [36]. Pregnant women are especially vulnerable to substance exposure, as drugs can cross the placental barrier, potentially causing preterm birth, birth defects, or long-term cognitive and behavioral issues [37]. For example, prenatal exposure to methamphetamine has been linked to impaired fetal brain development, low birth weight, and an increased risk of neurodevelopmental disorders [38]. Studying these potential development impairments in *C. elegans* provides insights into potential deleterious effects that could be relevant to both adolescent brain and fetal development. *C. elegans* undergoes rapid, regulated growth through four larval stages (L1 to L4), developing body plan, internal organs, and reproductive structures. Upon reaching the L4 stage, the organism molts one final time to become an adult capable of reproduction. This well-orchestrated progression occurs within a few days under optimal conditions [39], but can be affected by several factors, including exposure to heavy metals, pesticides, and drugs [40,41,42].

Based on the toxicity results, sublethal concentrations of 2C-I and 25I-NBOMe were selected to evaluate the potential effects of these drugs on animal development and reproduction. Our results clearly demonstrate that sublethal concentrations of 2C-I and 25I-NBOMe prompt important development changes in *C. elegans*, with both drugs significantly decreasing the average length of the worms. From these findings, we can speculate that exposure to drugs like 2C-I and 25I-NBOMe may disrupt developmental processes in humans similar to what was observed in *C. elegans*. This is particularly concerning given the user population, with young adults, as previously mentioned, representing a significant portion of consumers. Knowing that this group is in important stages of neurodevelopment, they may be especially predisposed to the adverse effects of these drugs [34,35,36]. Similarly, piperazine-designer drugs also caused developmental alterations in *C. elegans*. A significant and concentration-dependent reduction in the body area was observed following exposure to BZP (2.5–65.0 mM), MeOPP (0.5–7.5 mM), and MDBP (0.25–2.5 mM) for 30 min [41].

Apart from their effects on development, these substances can have significant impacts on reproductive health. Indeed, drugs such as methamphetamine are known to have significant impacts on reproductive health, including effects on sperm function, embryo development, and newborns [43]. Although this risk is recognized, evidence linking NPSs to reproductive health remains limited. With infertility on the rise globally [44], understanding the effects of drugs on fertility is vital. Given the challenges of obtaining data from human studies, alternative models are essential for exploring these parameters. *C. elegans* primarily reproduce as hermaphrodites, which can self-fertilize or mate with males. During their 2–3 weeks lifespan, hermaphrodites usually lay 300 to 350 eggs, with their rate and timing of egg-laying influenced by various environmental factors [39]. The vulva is a vital structure of the reproductive system in *C. elegans*, facilitating both copulation and egg-laying through its direct connection to the uterus. Mutations affecting vulva development can lead to egg-laying issues or morphological defects such as a protruding vulva or abnormal vulva eversion [45]. Thus, disruptions in vulval development can impact overall reproductive success.

Our results demonstrate that sublethal concentrations of 2C-I and 25I-NBOMe significantly decrease the total progeny of *C. elegans*. However, the percentage of hatched embryos did not show significant differences, indicating that these drugs do not affect progeny viability. Additionally, no visible defects were observed in the vulva of the worms treated with either drug. This suggests that the drugs do not directly alter the vulval structure or cause visible malformations, but subtle effects on vulval function or reproductive health cannot be ruled out. Although there are important differences in complexity and physiology between *C. elegans* and humans, these results provide valuable insights into potential risks for humans. Accordingly, research conducted by Harlow et al. showed that 89% of the compounds that negatively affected *C. elegans* egg viability were also associated with developmental impacts in mammals (ToxRef database) [28]. These findings indicate that even low-dose exposure to these substances could affect fertility. Given the rise in infertility and prevalent psychoactive drug use [44,46], these results highlight the need for further research to assess the impact of such substances on human reproductive health and inform public health strategies.

Other studies have also reported the effects of drugs on the reproductive health of *C. elegans*, further demonstrating the model’s suitability for evaluating the reproductive impacts of psychoactive substances. Recently, Yücel showed that ketamine (2.5 mM) reduced the average brood size in treated animals by 63% compared to the controls. They also observed that ketamine induces apical extracellular matrix modifications in the worms’ vulva and cuticle. Moreover, ketamine-treated worms had a 15% slower pharyngeal pumping rate, indicating that impaired feeding could be linked with the observed developmental delays—only 18% of the ketamine-treated animals reached adulthood 54 h post-L1 stage, compared to 100% of the control group [47]. Moreover, exposure for 30 min to piperazine designer drugs significantly reduced the nematodes brood size (eggs and larval worms) at the higher concentrations tested of BZP (65 mM) and MeOPP (5–7.5 mM), while MDBP (0.5–2 mM) had no significant effect [41]. Overall, both ketamine and piperazine derivatives significantly reduced brood size in *C. elegans*, indicating their potential to impair fertility and reproduction, and underscoring the importance of whole-organism models like *C. elegans* in toxicology research.

Regarding potential mechanisms enrolled in the observed 2C-I and 25I-NBOMe effects on development and reproductive systems in *C. elegans*, we highlight that designer drugs are known to disturb neurotransmission by altering the release, uptake, and/or receptor binding of neurotransmitters, leading to significant changes in brain function and behavior [48]. 2C-I and 25I-NBOMe are thought to primarily interact with serotonin receptors, particularly the 5-HT_2A_ subtype, where they seem to act as agonists [20,49]. However, interactions with other neurotransmitters should not be overlooked. Accordingly, beyond increased serotonin levels, 25I-NBOMe has also been found to increase extracellular dopamine and glutamate in the rat’s frontal cortex [7]. In *C. elegans*, neurotransmitters such as dopamine and serotonin affect pharyngeal pumping (the mechanism by which the worm ingests food), locomotion, learning, and egg-laying [39]. Indeed, Nagashima and Oami et al. reported that dopamine negatively regulates the body size in *C. elegans* [50]. Schreiber and McIntire showed that a 1-h exposure to methamphetamine (8–16 mM) and MDMA (0.5–10 mM) significantly decreased *C. elegans* egg-laying, while also inhibiting feeding (methamphetamine: 2–16 mM and MDMA: 0.5–10 mM) and causing paralysis (methamphetamine: 16 mM and MDMA: 10 mM) [51]. Therefore, given our results, it is possible that 2C-I and 25I-NBOMe affect *C. elegans* body size and egg-laying by interfering with neurotransmitter signaling, potentially with dopaminergic and serotoninergic signaling, although this relationship requires further investigation.

## 4. Materials and Methods

### 4.1. Chemicals

The 2C-I and 25I-NBOMe were synthesized and fully characterized using mass spectrometry (MS) and nuclear magnetic resonance (NMR) techniques, as previously described by [21]. Drug stock solutions were prepared in DMSO and stored at −20 °C. On the day of the experiments, working solutions were freshly diluted in phosphate-buffered saline (PBS).

Tryptone/Peptone ex casein (8952.1) was acquired from Carl Roth (Karlsruhe, Germany). Agar granulated (MB02903), LB broth (MB14502), and Agarose (MB02703) were acquired from NZYtech (Lisbon, Portugal). Calcium chloride dihydrate (CaCl_2_·2H_2_O, 223506), magnesium sulfate heptahydrate (MgSO_4_·7H_2_O, 230391), sodium hydrogen phosphate (S3264), cholesterol (C8667), and sodium azide (S8032) were acquired from Sigma-Aldrich (Darmstadt, Germany). Sodium chloride (NaCl, 31434) was acquired from Honeywell (Brussels, Belgium). Potassium di-hydrogen phosphate (KH_2_PO_4_, 131509.1211) was acquired from ITW Reagents (Barcelona, Spain).

### 4.2. C. elegans Maintenance

The DC19 *C. elegans* strain was obtained from Caenorhabditis Genetics Center (University of Minnesota, Minneapolis, MN, USA). This strain carries a mutation in the *bus-5* gene, specifically the *bus-5(br19)* allele. The *bus-5* gene is involved in the synthesis of the cuticle, the protective outer layer of the worm. Mutations in *bus-5* can compromise the integrity and function of the cuticle, leading to increased susceptibility to environmental stresses and pathogens. This makes the DC19 strain a valuable model for toxicity assessment [27].

The strain was maintained at 22 °C in nematode growth medium (NGM) plates seeded with OP50 bacteria. NGM plates were prepared by dissolving bacto-agar (1.7% *w*/*v*), NaCl (0.3% *w*/*v*), and bacto-peptone (0.25% *w*/*v*) in distilled H_2_O, followed by autoclaving at 110 °C for 30 min. After autoclaving, the medium was supplemented with 1 mM CaCl_2_·2H_2_O, 1 mM MgSO_4_·7H_2_O, 25 mM KH_2_PO_4_ buffer (pH 6), and 5 µg/mL cholesterol. The supplemented medium was then dispensed into Petri dishes (Corning, BP53–06) and allowed to solidify at room temperature.

OP50 was prepared by thawing a glycerol stock and inoculating it into autoclaved (121 °C, 15 min) liquid LB broth (2.5% *w/v* in distilled H_2_O), followed by overnight incubation, at 37 °C, with shaking (200 rpm). Once the culture became turbid, indicating bacterial growth, the culture was divided by several Falcon tubes and kept at 4 °C until use. After the NGM plates had dried (~1 week) at room temperature, 250 µL of OP50 bacterial suspension was added to each NGM plate, as a food source, and the plates were stored at 4 °C until use.

### 4.3. Synchronization at the L1 Stage

Adult worms of DC19 *C. elegans* were synchronized at the L1 stage (synchronized L1s) by washing three times with M9 buffer (86 mM NaCl, 42 mM Na_2_HPO_4_, 22 mM KH_2_PO_4_, and 1 mM MgSO_4_·7H_2_O in distilled H_2_O) and pelleted, at 800× *g* for 3 min each time. Then, a bleaching solution (6:2:1 of 1 M NaCl, household bleach, and 5 M NaOH) was added, and the worm suspension was vortexed for 4 min. Embryos were pelleted by centrifugation, at 800× *g* for 3 min, and washed three times with M9 buffer. Embryos were allowed to hatch in the M9 buffer overnight at room temperature.

### 4.4. Drugs Exposure

Synchronized L1s were exposed to 2C-I (0.1–10 mM) and 25I-NBOMe (0.05–0.5 mM) in the M9 buffer containing OP50, in 48-well plates, for 24 h at 22 °C. After the exposure period, the worms were transferred to NGM plates with OP50, and the total number of alive and dead worms was counted. Then, the percentage of surviving animals was determined by calculating the ratio of dead worms to the total number of worms. Drugs were tested at least in five independent experiments, with an average of 160 worms counted per condition, including both alive and dead worms.

### 4.5. Animal Length

Synchronized L1s were exposed to 2C-I (0.0, 0.5, and 1.0) and 25I-NBOMe (0.00, 0.05, and 0.10) in the M9 buffer containing OP50, in 48-well plates, for 24 h at 22 °C. Following drug exposure, the animals were transferred and left to grow in NGM plates with OP50, at 22 °C. Twenty-four and forty-eight hours after drug exposure, approximately 20 alive worms per condition were imaged using a Nikon SMZ745T stereoscope equipped with MD-E3ISPM-E3 8.3 camera controlled by ToupView software (ToupTek Photonics, Hangzhou, ZJ, China). The animal length was measured by tracing a segmented line along the animal body using ImageJ software (version 1.53k, Bethesda, MD, USA). Drugs were tested at least in five independent experiments, with a minimum of 100 alive worms measured per condition.

### 4.6. Embryonic Viability and Brood Size

Synchronized L1s were exposed to 2C-I (0.0, 0.5, and 1.0 mM) and 25I-NBOMe (0.00, 0.05, and 0.10 mM) in the M9 buffer containing OP50, in 48-well plates, for 24 h at 22 °C. Following drug exposure, the worms were transferred to NGM plates with OP50 and allowed to grow for an additional period of 48 h, at 22 °C. Subsequently, five young adult animals (F0 generation) were singled out to new NGM plates with OP50 and allowed to lay embryos for 48 h, at 22 °C. Then, the parent animals (F0) were removed from the plates, and the total F1 progeny (embryos and larvae) was counted 24 h later. The total number of progeny was calculated for each condition, and the percentage of hatched embryos was calculated by the ratio of hatched embryos to the total number of progeny. Drugs were tested at least in three independent experiments, with the brood size analyzed in at least 15 adult worms.

### 4.7. Differential Interference Contrast (DIC) Microscopy

Synchronized L1s were exposed to 2C-I (0.0 and 1.0 mM) and 25I-NBOMe (0.0 and 0.10 mM) in the M9 buffer containing OP50, in 48-well plates, for 24 h at 22 °C. Following drug exposure, the worms were transferred and left to grow in NGM plates with OP50, at 22 °C. Forty-eight hours after drug exposure, the worms were mounted on freshly prepared 2% (*w*/*v*) agarose pads, paralyzed with sodium azide (50 mM), and covered with a coverslip. DIC imaging was performed on an Axio Observer Z1 microscope (Zeiss) equipped with a CSU-X1 confocal scanner (Yokogawa Electric Corporation), an HXP 120C Illuminator, and controlled by AxioVision software (Zeiss). Images were acquired at 1 × 1 binning using a 63× NA 1.4 Plan-Apochromat objective (Zeiss), to conduct a detailed morphological examination of the vulval region, assessing for potential abnormalities.

### 4.8. Statistical Analysis

Statistical analyses were performed using GraphPad Prism^®^ software, version 9.0 (San Diego, CA, USA). Concentration-response curves were obtained using the least squares fitting method, and comparisons between curves (LogEC_50_, Top, Bottom, and Hill Slope) were conducted using the extra sum-of-squares F test. Statistical differences were determined using the One-Way ANOVA on ranks (Kruskal–Wallis test), followed by Dunn’s multiple comparisons *post-hoc* test, with *p* < 0.05 considered significant.

## 5. Conclusions

2C-I and 25I-NBOMe significantly impacted the survival rate of *C. elegans* in a concentration-dependent manner, with 25I-NBOMe exhibiting approximately six times greater toxicity than 2C-I. This higher toxicity correlates well with the 25I-NBOMe’s higher lipophilicity, which enhances its ability to permeate biological membranes, thereby increasing its overall toxic effect. Moreover, the observation that sublethal concentrations of 2C-I and 25I-NBOMe significantly impact development and reproduction in *C. elegans* underscores the potential risks associated with low-dose exposure, especially in vulnerable populations. This is the first study to demonstrate the toxic effects of 2C-I and 25I-NBOMe in *C. elegans*, establishing it as a valuable model for assessing the developmental and reproductive impacts associated with the consumption of drugs.

## Figures and Tables

**Figure 1 ijms-26-03039-f001:**
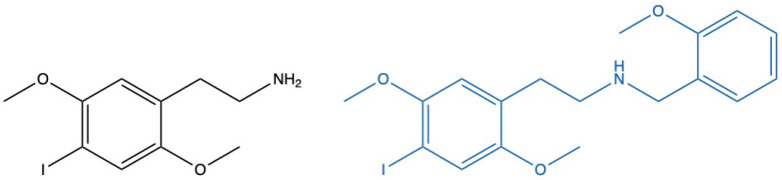
Chemical structure of 4-iodo-2,5-dimethoxyphenethylamine (2C-I, **left**) and 2-(4-iodo-2,5-dimethoxyphenyl)-*N*-(2-methoxybenzyl)ethanamine (25I-NBOMe, **right**).

**Figure 2 ijms-26-03039-f002:**
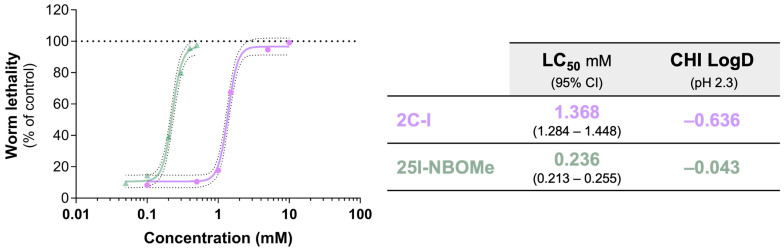
Concentration-response curves of 2C-I (0.0–10 mM) and 25I-NBOMe (0.0–0.5 mM) in *C. elegans*, 24 h after exposure. The least squares method was used to define the lines of best fit for our data set. Results are presented as mean with a 95% Confidence Interval (CI), with dashed lines representing the upper and lower limits. The concentration of drugs necessary to reach half of the maximum effect—in this case, worm lethality (LC_50_)—is also presented. On average, 160 worms were counted per condition, including both living and dead worms. At least five independent experiments were performed. The chromatographic hydrophobicity index (CHI) values were obtained from [21]. According to their toxic profile, the following concentrations were selected for the subsequent experiments: 0.5 and 1.0 mM for 2C-I and 0.05 and 0.10 mM for 25I-NBOMe. The selected concentrations correspond to survival rates of 89.51 and 83.56% for 2C-I at 0.5 mM and 1.0 mM, respectively, and 90.44 and 85.51% for 25I-NBOMe at 0.05 mM and 0.10 mM, respectively, 24 h after exposure (Table 1).

**Figure 3 ijms-26-03039-f003:**
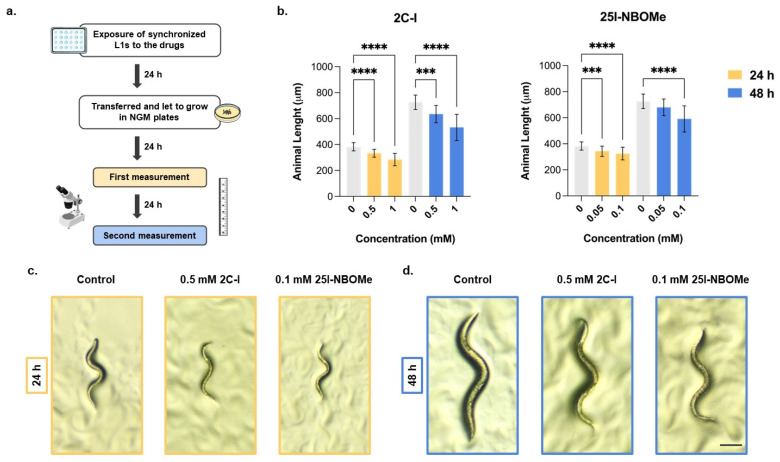
(**a**) Schematic representation of the procedure for measuring animal length, and (**b**) animal length measured 24 (yellow) and 48 (blue) hours after a 24-h exposure to 2C-I (0.0, 0.5 and 1 mM) or 25I-NBOMe (0.00, 0.05 and 0.1 mM) drugs, in comparison with control animals (grey). Values are expressed as mean ± SD from at least five independent experiments, with a minimum of 100 alive worms measured per condition. Statistical comparisons were obtained using a Kruskal–Wallis test, followed by Dunn’s multiple comparisons *post hoc* test, *** *p* < 0.001; **** *p* < 0.0001 drugs vs. control (0 mM). (**c**,**d**) Representative bright-field images of control animals (0 mM) and animals exposed for 24 h to sublethal concentrations of 2C-I (0.5 mM) or 25I-NBOMe (0.1 mM) and then let to grow in NGM-seeded plates for an additional period of 24 (**c**) or 48 (**d**) hours after drug exposure. The scale bar represents 100 µm.

**Figure 4 ijms-26-03039-f004:**
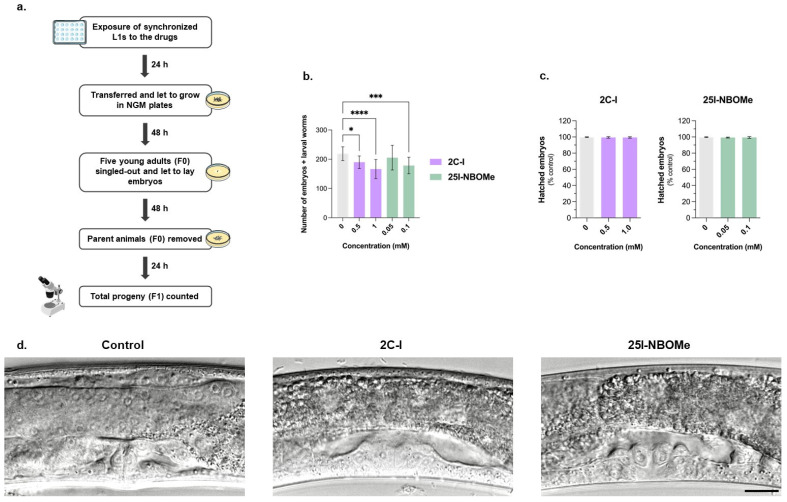
(**a**) Schematic representation of the procedure for assessing embryonic viability and brood size. (**b**) Brood size (number of embryos and larvae) of control animals (grey) and animals exposed for 24 h to 2C-I (0.0, 0.5, and 1 mM, purple) or 25I-NBOMe (0.00, 0.05, and 0.1 mM, green) drugs, and then left to grow for an aSdditional period of 48 h. Values are expressed as mean ± SD from three independent experiments. Statistical comparisons were obtained using a Kruskal–Wallis test, followed by Dunn’s multiple comparisons *post-hoc* test. * *p* < 0.05; *** *p* < 0.001; **** *p* < 0.0001; drugs vs. control (0 mM). (**c**) Viability of F1 generation after exposing F0 animals to 2C-I (0.0, 0.5, and 1 mM) or 25I-NBOMe (0.00, 0.05, and 0.1 mM) drugs for 24 h exposure. Values are expressed as mean ± SD from three independent experiments, with the brood size analyzed in at least 15 adult worms. Statistical comparisons were obtained using a Kruskal–Wallis test, followed by Dunn’s multiple comparisons *post-hoc* test. (**d**) DIC images of *C. elegans* vulva under control conditions (0 mM) and after exposure to 2C-I or 25I-NBOMe. The worms were exposed for 24 h to 2C-I (1.0 mM) or 25I-NBOMe (0.1 mM) and left to grow in NGM-seeded plates for an additional period of 48 h after drug exposure. The scale bar represents 20 µm.

**Table 1 ijms-26-03039-t001:** Survival rates (%) of *C. elegans* exposed to different concentrations of 2C-I (0.5 and 1.0 mM) or 25I-NBOMe (0.05 mM and 0.10 mM) drugs for 24 h. [** *p* < 0.01 Control vs. 2C-I or 25I-NBOMe].

Control	2C-I		25I-NBOMe	
0 mM	0.5 mM	1.0 mM	0.05 mM	0.10 mM
Mean survival rate (%)				
95.83	89.51	82.53 **	90.44	85.51

## Data Availability

The data that support the findings of this study are available from the corresponding author upon request.

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
