# Peer review of "Sublethal Concentrations of 2C-I and 25I-NBOMe Designer Drugs Impact *Caenorhabditis elegans* Development and Reproductive Behavior"

_ijms, 2025, doi:10.3390/ijms26073039_

Round 1

Reviewer 1 Report

Comments and Suggestions for Authors

The manuscript titled “Sublethal Concentrations of 2C-I and 25I-NBOMe Designer Drugs Impact Caenorhabditis elegans Development and Reproductive Behavior” investigates the toxicity of these synthetic psychoactive substances using C. elegans as an experimental model. The study evaluates their effects on animal length, embryonic viability, and brood size across different concentrations, revealing a clear dose-dependent toxicity, with 25I-NBOMe being significantly more toxic than 2C-I. While the findings are valuable, and I acknowledge the authors' effort, I believe the study would benefit from additional biological assays to provide a more comprehensive understanding of the compounds' effects. In its current form, the manuscript presents limited results, making it unsuitable for publication in a high-impact journal like the International Journal of Molecular Sciences.

Nevertheless, I have several comments that could help improve the manuscript:

-              Why was the exposure conducted at the L1 stage rather than from eggs or another developmental stage?

-              Line 24: C. elegans should be in italics.

-              How can these findings be extrapolated to potential toxic effects in humans, particularly regarding their use as recreational drugs? The three assays performed provide only a basic assessment of C. elegans toxicity. It might be beneficial to evaluate additional toxicity endpoints.

-              The lethality range is quite narrow; does this imply a similarly narrow dosage window for human use?

-              How many worms were used for the lethality curves? The manuscript states that at least five replicates were performed, but it does not specify the number of worms per replicate.

-              Following the lethality test, the authors selected concentrations that already exhibit a certain percentage of lethality. Would it not be more appropriate to select concentrations with 100% survival? What was the rationale behind these choices? Furthermore, among the four concentrations tested (two for each compound), only one shows a statistically significant difference from the control.

-              In the growth assay, worms were exposed from the L1 stage to the treatment for 24 hours, then left for an additional 24 or 48 hours before measuring their length. Why was this approach chosen? Would it not be more appropriate to expose the worms for the entire duration of the experiment? A 24-hour exposure period may be insufficient, and maintaining the treatment throughout the experiment could provide more meaningful insights.

-              The treatments negatively affect growth, resulting in smaller worms. What are the potential biological implications of this finding?

Investigating the underlying mechanisms might strengthen the study.

-              How many individuals and replicates were included in the growth assay?

This information should be explicitly stated, and the sample size should also be indicated in the figure panels. Similarly, the number of animals analyzed in the lethality experiment should be clearly reported in the Methods section.

-              For example, the number of animals analyzed in the lethality experiment is also missing.

-              In the reproductive assay, using only five animals per replicate seems

insufficient. Increasing the sample size would improve statistical robustness.

-              Figure 4.D. Scale bar or magnification reference should be included.

-              The discussion is somewhat lengthy and tedious, containing information that could be summarized, particularly up to line 240.

-              Why was the DC19 strain used instead of the N2 wild-type strain for the toxicity assessment?

Author Response

Reviewer 1

The manuscript titled “Sublethal Concentrations of 2C-I and 25I-NBOMe Designer Drugs Impact Caenorhabditis elegans Development and Reproductive Behavior” investigates the toxicity of these synthetic psychoactive substances using C. elegans as an experimental model. The study evaluates their effects on animal length, embryonic viability, and brood size across different concentrations, revealing a clear dose-dependent toxicity, with 25I-NBOMe being significantly more toxic than 2C-I. While the findings are valuable, and I acknowledge the authors' effort, I believe the study would benefit from additional biological assays to provide a more comprehensive understanding of the compounds' effects. In its current form, the manuscript presents limited results, making it unsuitable for publication in a high-impact journal like the International Journal of Molecular Sciences.

We sincerely thank the Reviewer for its thoughtful evaluation of our manuscript and for recognizing the value of our findings. We appreciate the recognition of the dose-dependent toxicity of 2C-I and 25I-NBOMe revealed in our study, as well as the effort we have put into this work. We would like to highlight that, to the best of our knowledge, this study is the first to investigate the toxicity of these designer drugs using C. elegans as an experimental model. While we agree that further studies are necessary to provide a more comprehensive understanding of the molecular and cellular pathways underlying the observed effects, our primary aim was to gain initial insights into the toxicity of these substances and validate the usefulness of C. elegans as a model organism for this type of research. The results presented here lay important groundwork for future investigations, which will delve deeper into the specific mechanisms of action and signaling pathways affected by these compounds. We believe this initial exploration provides a solid foundation for advancing our understanding of the toxicological profiles of these designer drugs, while demonstrating the potential of C. elegans as a valuable tool in this field. We hope this clarification emphasizes the novelty and significance of our work, even within its current scope, and demonstrates its potential contribution to future research efforts. 

Nevertheless, I have several comments that could help improve the manuscript:

  1. Why was the exposure conducted at the L1 stage rather than from eggs or another developmental stage?

We acknowledge the Reviewer's question. Through the synchronization process, which involves bleaching adult worms to obtain embryos, we generate a homogeneous population of C. elegans L1 larvae. On the other hand, distinguishing between dead and alive embryos is challenging. This adds another layer of uncertainty, as variations in the frequency of dead embryos across different experiments could lead to increased variability in the number of individuals used for drug exposure. Additionally, achieving a synchronized population at stages beyond L1 is nearly impossible in C. elegans. Although starting experiments with embryos would be possible, it is technically more challenging and, since embryos hatch at slightly different times, it would result in developmental variability. In contrast, L1 larvae can be effectively synchronized by allowing them to hatch in the absence of food, which keeps them in the L1 stage until food is present. Therefore, synchronizing to obtain a homogeneous population of L1s is the most effective method for minimizing variability, leading to more precise assessments of the drug's effects.

  1. Line 24: C. elegans should be in italics.

We appreciate the Reviewer's comment, and we will ensure that C. elegans is properly italicized in the final version of the manuscript.

  1. How can these findings be extrapolated to potential toxic effects in humans, particularly regarding their use as recreational drugs? The three assays performed provide only a basic assessment of C. elegans toxicity. It might be beneficial to evaluate additional toxicity endpoints.

We acknowledge the Reviewer's question. To the best of our knowledge, this is the first study investigating the toxicity of phenethylamine designer drugs using C. elegans as a model. As an initial approach, we focused on the general toxicity assessment of 2C-I and 25I-NBOMe, evaluating key endpoints such as lethality (including LC50 estimation), developmental impairment, and reproductive behavior to gain fundamental insights into their toxicological effects. C. elegans has extensive homology to mammalian genes and well-studied developmental and reproductive systems, serving as a valuable tool for toxicity assessments with direct relevance to human biology. Although we acknowledge that direct extrapolation to human toxicity is challenging, this is a common limitation when using animal models. Differences in metabolism, drug absorption, distribution, and excretion between species may result in variations in toxic effects. Additionally, complex physiological interactions present in humans may not be fully replicated in simpler model organisms. However, C. elegans provides a cost-effective and well-established system for identifying potential toxic effects and underlying mechanisms, offering valuable preliminary data to guide further studies in higher organisms. Future studies could focus on a more detailed mechanistic assessment of toxicity, including oxidative stress responses, mitochondrial dysfunction, or alterations in neurotransmission pathways. Investigating these molecular and cellular effects would provide a deeper understanding of how these substances interact with biological systems and could help bridge the gap between C. elegans findings and potential human toxicity outcomes.

  1. The lethality range is quite narrow; does this imply a similarly narrow dosage window for human use?

We appreciate the Reviewer's question. While the lethality range for 2C-I and 25I-NBOMe in C. elegans is narrow, direct extrapolation to human dosage window is complex due to significant differences in physiology, metabolism, and drug distribution between species. Additionally, factors such as route of administration and individual variability further complicate direct comparisons to human use. However, user reports suggest that a common oral dose of 2C-I ranges from 10 to 25 mg, with a strong dose typically falling between 25 and 30 mg. For 25I-NBOMe, a common sublingual dose is reported to be between 500 and 750 µg, while a strong dose ranges from 750 µg to 1 mg. These reports indicate a relatively small margin between typical and strong doses, aligning with concerns regarding the potential for severe adverse effects. While the findings in C. elegans cannot be directly extrapolated to humans, the observed narrow lethality range may reflect the limited safety margin associated with these substances. Further studies using mammalian models and human-relevant systems would be necessary to better characterize their pharmacological and toxicological profiles.

  1. How many worms were used for the lethality curves? The manuscript states that at least five replicates were performed, but it does not specify the number of worms per replicate.

We appreciate the Reviewer's question. On average, around 160 total worms in each independent experiment (live and dead) were counted per condition for the lethality curves, across at least five independent experiments. This information was incorporated in the final version of the manuscript to ensure transparency regarding the experimental details.

  1. Following the lethality test, the authors selected concentrations that already exhibit a certain percentage of lethality. Would it not be more appropriate to select concentrations with 100% survival? What was the rationale behind these choices? Furthermore, among the four concentrations tested (two for each compound), only one shows a statistically significant difference from the control.

We appreciate the Reviewer’s comment. Based on the results of the lethality assay, we selected concentrations for the subsequent experiments – development impairment and reproductive behavior – that would allow us to observe measurable effects without overwhelming the model. For 2C-I, we chose 0.5 and 1 mM, and for 25I-NBOMe, we selected 0.05 and 0.1 mM. Among these, only the 1 mM concentration of 2C-I caused a significant reduction in animal survival, with a decrease of approximately 10% compared to the control. The other concentrations tested were considered sublethal, showing no significant differences compared to the control, as indicated in Table 1 of the manuscript. The rationale behind this approach was to identify concentrations that would allow us to detect the toxic effects of the compounds while avoiding excessive toxicity that could result in non-physiological outcomes or complete lethality. By using sublethal concentrations, we aimed to investigate potential sublethal effects and gain a more comprehensive understanding of the broader impact of these drugs.

  1. In the growth assay, worms were exposed from the L1 stage to the treatment for 24 hours, then left for an additional 24 or 48 hours before measuring their length. Why was this approach chosen? Would it not be more appropriate to expose the worms for the entire duration of the experiment? A 24-hour exposure period may be insufficient, and maintaining the treatment throughout the experiment could provide more meaningful insights.

We appreciate the Reviewer's question. The 24-hour exposure period was chosen because significant effects were already observed within this timeframe, indicating that prolonged exposure may not be necessary to evaluate the impact of the drugs. We measured the worm length following 24- and 48-hours post the 24h-exposure to better distinguish growth differences. Immediately after exposure, the worms were smaller and more challenging to measure accurately and would result in less clear differences. By allowing more time for growth, we were capable of obtaining more reliable and distinguishable measurements. Moreover, the stability of the drugs in the medium is likely to reduce over time, meaning that extending the exposure might not proportionally increase the toxic effects. By allowing worms to recover after the initial exposure, we could also evaluate whether the exposure had lasting effects on growth rather than acute toxicity. Since significant effects were observed for the selected concentrations and time points, longer incubation periods were deemed unnecessary. Furthermore, the selected exposure duration reflects better the human consumption patterns, where drug use is often episodic rather than continuous, making extended exposure less relevant in this context

  1. The treatments negatively affect growth, resulting in smaller worms. What are the potential biological implications of this finding? Investigating the underlying mechanisms might strengthen the study.

We acknowledge the Reviewer’s comment. The observed reduction in worm size suggests that 2C-I and 25I-NBOMe interfere with normal developmental processes, which could be attributed to disruptions in energy metabolism, feeding behavior, or cellular signaling pathways. Given that growth impairment is often linked to broader physiological stress responses, this finding may indicate an overall toxic burden on the organism. While our study primarily aimed to provide a general assessment of toxicity, future research could focus on elucidating the underlying mechanisms. Potential avenues of investigation include assessing mitochondrial function and oxidative stress, evaluating changes in nutrient uptake and metabolism, or examining alterations in key developmental signaling pathways. These mechanistic insights could further clarify how these substances impact biological systems and enhance the relevance of our findings.

How many individuals and replicates were included in the growth assay? This information should be explicitly stated, and the sample size should also be indicated in the figure panels. Similarly, the number of animals analyzed in the lethality experiment should be clearly reported in the Methods section. For example, the number of animals analyzed in the lethality experiment is also missing. In the reproductive assay, using only five animals per replicate seems insufficient. Increasing the sample size would improve statistical robustness.

We appreciate the Reviewer's concern and acknowledge the importance of clearly reporting the sample sizes. For the lethality assay, we counted on average 160 worms, both alive and dead, per condition and performed at least 5 independent experiments. For the growth assay, we measured the size at least 100 live worms per condition from at least 5 independent experiments. Additionally, we analyzed the brood size of at least 15 adult worms. We recognize the importance of statistical robustness and believe that our sample sizes provide sufficient power to detect meaningful differences. Nonetheless, we will ensure that this information is explicitly stated in the Methods section and presented in the figure legends.

  1. Figure 4.D. Scale bar or magnification reference should be included.

We appreciate the Reviewer's insightful suggestion. In the final version of the manuscript, we incorporated a scale bar directly into Figure 4.D. Additionally, we would like to note that the scale bar information was also included in the figure legend for further clarity.

  1. The discussion is somewhat lengthy and tedious, containing information that could be summarized, particularly up to line 240.

We appreciate the Reviewer thoughtful feedback regarding the length and detail of the discussion. In response, we have carefully revised this section, streamlining the content to improve readability and conciseness while maintaining the necessary depth and clarity. 

  1. Why was the DC19 strain used instead of the N2 wild-type strain for the toxicity assessment?

We appreciate the Reviewer’s question. The DC19 strain was selected for this toxicity assessment because it carries a mutation in the bus-5 gene, which is involved in the synthesis of the cuticle, the protective outer layer of the worm. This mutation compromises the integrity and function of the cuticle, making the DC19 strain more susceptible to environmental stresses and xenobiotics. Since the C. elegans cuticle has been reported to act as a barrier to chemical uptake, it may lead to an underestimation of toxic effects [1], Thus, the use of the DC19 strain helps ensure that toxic effects are not underestimated.

Reference

  1. Xiong, H., C. Pears, and A. Woollard, An enhanced C. elegans based platform for toxicity assessment. Sci. Rep., 2017. 7(1): p. 9839.

Reviewer 2 Report

Comments and Suggestions for Authors

Gil-Martins et al. conducted a study on the developmental and reproductive toxicity of the psychoactive substances 2C-I and 25I-NBOMe using the C. elegans model. The manuscript is well-structured, and the experimental design appears to be carefully planned and executed. The findings contribute valuable insights into the toxicological effects of these substances, highlighting their potential impact on biological development and reproduction. However, to further enhance the quality and clarity of the manuscript, I suggest the authors consider the following improvements.

  1. When evaluating the toxicological effects of 2C-I and 25I-NBOMe, the authors assessed only three endpoints: (a) LC50, and (b) effects on development (body length), and (c) reproduction (brood size and embryonic lethality). This data alone is insufficient to fully support the conclusions. I strongly recommend that the authors include additional relevant data to strengthen their findings.
  2. Using C. elegans with the bus-5 mutation to study the toxicological profile of psychoactive substances (2C-I and 25I-NBOMe) is an interesting approach. However, have you screened the effects of 2C-I and 25I-NBOMe on wild-type (Bristol N2) C. elegans? If so, what were your observations, and why is this data not included in the manuscript? If not, what was the rationale for omitting such an important control? Given the significance of using wild-type worms in toxicological studies, I assume the authors are aware of its importance.
  3. Additionally, on what basis were the two different concentrations of 2C-I (0.5 and 1.0 mM) and 25I-NBOMe (0.05 and 0.10 mM) selected, considering their differing LC50 values?
  4. Finally, what specific observations were made in Figures 1c and 2d? Please clarify any noticeable changes in the figures and provide a more detailed explanation in the text.
  5. I suggest the authors include the number of replicates used for each independent trial.
  6. The binomial name should be italicized throughout the text (i.e., C. elegans).
  7. Additionally, I recommend maintaining consistency by using either "OP50" or "E. coli OP50" consistently throughout the manuscript.

Author Response

Reviewer 2

Gil-Martins et al. conducted a study on the developmental and reproductive toxicity of the psychoactive substances 2C-I and 25I-NBOMe using the C. elegans model. The manuscript is well-structured, and the experimental design appears to be carefully planned and executed. The findings contribute valuable insights into the toxicological effects of these substances, highlighting their potential impact on biological development and reproduction. However, to further enhance the quality and clarity of the manuscript, I suggest the authors consider the following improvements.

We sincerely thank the Reviewer for its thoughtful evaluation of our manuscript and for recognizing the value of our findings. We are grateful for the recognition of the manuscript structure, experimental design and its contribution to the field. We also appreciate the constructive feedback and will carefully consider the suggested improvements to enhance the revised manuscript.

  1. When evaluating the toxicological effects of 2C-I and 25I-NBOMe, the authors assessed only three endpoints: (a) LC50, and (b) effects on development (body length), and (c) reproduction (brood size and embryonic lethality). This data alone is insufficient to fully support the conclusions. I strongly recommend that the authors include additional relevant data to strengthen their findings.

We acknowledge the Reviewer's question. To the best of our knowledge, this is the first study investigating the toxicity of phenethylamine designer drugs using C. elegans as a model. As an initial approach, we focused on the general toxicity assessment of 2C-I and 25I-NBOMe, evaluating key endpoints such as lethality (including LC50 estimation), developmental impairment, and reproductive behavior to gain fundamental insights into their toxicological effects, and also to validate the usefulness of C. elegans as a model organism for this type of research. C. elegans has extensive homology to mammalian genes and well-studied developmental and reproductive systems, serving as a valuable tool for toxicity assessments with direct relevance to human biology. Although we acknowledge that direct extrapolation to human toxicity is challenging, this is a common limitation when using animal models. Differences in metabolism, drug absorption, distribution, and excretion between species may result in variations in toxic effects. Additionally, complex physiological interactions present in humans may not be fully replicated in simpler model organisms. However, C. elegans provides a cost-effective and well-established system for identifying potential toxic effects and underlying mechanisms, offering valuable preliminary data to guide further studies in higher organisms. Future studies could focus on a more detailed mechanistic assessment of toxicity, including oxidative stress responses, mitochondrial dysfunction, or alterations in neurotransmission pathways. Investigating these molecular and cellular effects would provide a deeper understanding of how these substances interact with biological systems and could help bridge the gap between C. elegans findings and potential human toxicity outcomes.

  1. Using C. elegans with the bus-5 mutation to study the toxicological profile of psychoactive substances (2C-I and 25I-NBOMe) is an interesting approach. However, have you screened the effects of 2C-I and 25I-NBOMe on wild-type (Bristol N2) C. elegans? If so, what were your observations, and why is this data not included in the manuscript? If not, what was the rationale for omitting such an important control? Given the significance of using wild-type worms in toxicological studies, I assume the authors are aware of its importance.

We appreciate the Reviewer’s comment. While we fully acknowledge the importance of using the wild-type N2 strain as a control in toxicological studies, our decision to focus on the DC19 strain was strategic and deliberate. The DC19 strain carries the bus-5(br19) mutation, which compromises cuticle integrity, leading to increased permeability. This is particularly relevant because the C. elegans cuticle is known to act as a barrier to chemical exposure, potentially leading to an underestimation of toxic effects. By using the DC19 strain, we aimed to create a more sensitive model for toxicity screening, ensuring more accurate detection of adverse effects. Indeed, previous studies have demonstrated that the enhanced permeability in bus-5 mutants allows for the observation of toxic effects at concentrations five to ten times lower than those required for wild-type worms [1]. This increased sensitivity is crucial for initial toxicity assessments, where detecting even subtle adverse effects can provide valuable insights. Nevertheless, future studies will include the N2 background to compare sensitivity differences and further validate our findings.

  1. Additionally, on what basis were the two different concentrations of 2C-I (0.5 and 1.0 mM) and 25I-NBOMe (0.05 and 0.10 mM) selected, considering their differing LC50 values?

We appreciate the Reviewer’s question. Given the significantly different LC50 values of the drugs – 1.368 mM for 2C-I and 0.236 mM for 25I-NBOMe – we selected concentrations that produced similar effects for both drugs (0.5 and 1.0 mM for 2C-I, and 0.05 and 0.10 mM for 25I-NBOMe) to allow direct comparisons. Based on the results of the initial lethality assays, only the 1.0 mM concentration of 2C-I caused a statistically significant reduction in worm viability, with a decrease of approximately 10% compared to the control. The other concentrations tested were considered sublethal, showing no significant differences from the control, as indicated in Table 1 of the manuscript. This approach was designed to identify concentrations that would allow us to detect sublethal toxic effects and gain a more comprehensive understanding of the broader impact of these drugs.

  1. Finally, what specific observations were made in Figures 1c and 2d? Please clarify any noticeable changes in the figures and provide a more detailed explanation in the text.

We appreciate the Reviewer's suggestion. In Figure 3C, we present representative images of non-exposed (control) and drug-exposed worms. We acknowledge that it might be challenging for readers to distinguish differences between the conditions. Therefore, we revised these panels to display a single animal representative of the effects illustrated in the graphs (differences in animal length upon drug exposure), making the panels clearer. Additionally, we provided a more detailed explanation in the text to better contextualize the meaning of these panels for the readers. 

  1. I suggest the authors include the number of replicates used for each independent trial.

We appreciate the Reviewer's concern and acknowledge the importance of clearly reporting the sample sizes. For the lethality assay, we counted on average 160 worms, both live and dead, per condition and performed at least five independent experiments. For the growth assay, we measured the size of approximately 100 live worms per condition from at least five independent experiments. Additionally, we analyzed the brood size of at least 15 adult worms. We will ensure that this information is explicitly stated in the Methods section and included in the relevant figure panels in the final version of the manuscript.

  1. The binomial name should be italicized throughout the text (i.e., C. elegans).

We appreciate the Reviewer's comment, and we will ensure that C. elegans is properly italicized in the revised version of the manuscript.

  1. Additionally, I recommend maintaining consistency by using either "OP50" or "E. coli OP50" consistently throughout the manuscript.

We acknowledge the Reviewer's recommendation and will ensure consistency by using "OP50” throughout the manuscript. We will make the necessary adjustments in the revised version to maintain uniform terminology.

Reference

[1] Lanzerstorfer, P., et al., Acute, reproductive, and developmental toxicity of essential oils assessed with alternative in vitro and in vivo systems. Arch Toxicol, 2021. 95(2): p. 673-691.

Reviewer 3 Report

Comments and Suggestions for Authors

The study by Gil-Martins et al., investigates the impact of two NPS 2C-I and 25I-NBOMe in the survival, development and reproduction of C. elegans. The study is well-designed, and each section of the manuscript is thoroughly developed. All results are well-discussed. However, the manuscript contains several grammatical, word choice, and sentence structure errors that need to be corrected such as "live vs dead". 

Comments on the Quality of English Language

the manuscript contains several grammatical, word choice, and sentence structure errors that need to be corrected such as "live vs dead". 

Author Response

Reviewer 3

The study by Gil-Martins et al., investigates the impact of two NPS 2C-I and 25I-NBOMe in the survival, development and reproduction of C. elegans. The study is well-designed, and each section of the manuscript is thoroughly developed. All results are well-discussed. However, the manuscript contains several grammatical, word choice, and sentence structure errors that need to be corrected such as "live vs dead".

We sincerely thank the Reviewer for its thoughtful evaluation of our manuscript and for recognizing the value of our findings. We are grateful for the recognition of the manuscript experimental design and its contribution to the field. We also appreciate the constructive feedback and will carefully consider the suggested improvements to enhance the revised manuscript.

  1. Comments on the Quality of English Language - the manuscript contains several grammatical, word choice, and sentence structure errors that need to be corrected such as "live vs dead".

We acknowledge the Reviewer's concern regarding the language quality. A thorough English revision has been conducted to improve grammar, word choice, and sentence structure throughout the manuscript. In particular, we have replaced the terms "live vs dead" with "alive and dead worms" to enhance clarity and accuracy.

Round 2

Reviewer 2 Report

Comments and Suggestions for Authors

I appreciate the authors for considering my comments and concerns to improve the manuscript. All the best.